# In Vitro Efficacy of Antibiotic Combinations with Carbapenems and Other Agents against Anaerobic Bacteria

**DOI:** 10.3390/antibiotics11030292

**Published:** 2022-02-22

**Authors:** Takumi Umemura, Mao Hagihara, Takeshi Mori, Hiroshige Mikamo

**Affiliations:** 1Department of Clinical Infectious Diseases, Aichi Medical University, Nagakute 480-1195, Japan; umemuratakumi@gmail.com (T.U.); hagimao@aichi-med-u.ac.jp (M.H.); mori99@aichi-med-u.ac.jp (T.M.); 2Department of Molecular Epidemiology and Biomedical Sciences, Aichi Medical University, Nagakute 480-1195, Japan

**Keywords:** carbapenem, clindamycin, minocycline, *Bacteroides*, *Peptostreptococcus*, combination therapy, checkerboard assay

## Abstract

We investigated the in vitro efficacy of combinations of carbapenems with clindamycin (CLDM) and minocycline (MINO) against *Bacteroides fragilis* and *Peptostreptococcus* species. We selected the carbapenems imipenem, meropenem, panipenem, doripenem, and biapenem. To evaluate the antibiotic efficacy of these combination regimens, the fractional inhibitory concentration index (FICI) was calculated against clinical isolates. Consequently, combination regimens of each carbapenem with CLDM or MINO showed synergistic or additive effects against 83.3–100.0% and no antagonistic effects against *P. anaerobius* isolates. However, against the *B. fragilis* group (*B. fragilis*, *B. thetaiotaomicron*, and *Parabacteroides distasonis*), although the combination with other carbapenems and CLDM or MINO did not show remarkable synergistic effects, the combination regimen of IPM with CLDM or MINO indicated mainly additive antibiotic efficacies (FICIs: >0.5 to ≤1.0) to *B. fragilis* groups. Then, antagonistic effects were admitted in only 5.6% of *B. fragilis* groups. The effectiveness of antibiotic combination therapy against pathogenic anaerobes has remained unclear. Then, our results can provide new insights to explore the effective combination regimens against multidrug-resistant anaerobic bacteria as empirical and definitive therapies, while this study used only carbapenem susceptible isolates. Hence, further studies are needed to use highly antibiotic-resistant anaerobic isolates to carbapenems.

## 1. Introduction

The overall number of immunocompromised patients with various disease types has been increasing in recent years with highly advanced medical treatments. This phenomenon resulted in an increase in the number of anaerobic bacteria isolated from patients, especially *Bacteroides* species [1,2]. Immune insufficiency can cause opportunistic infections, including infections with anaerobes and, then, underlying diseases (e.g., hemodialysis, malignancy and diabetes) are risk factors for anaerobic bacteremia [1]. The presence of anaerobes is known to be associated with a higher rate of mortality, even in polymicrobial infections [3,4,5,6]. There have been reports of increased multidrug-resistant *Bacteroides fragilis* group species, and *Peptostreptococcus anaerobius* showed the progression of antibiotic resistance worldwide over the last decade [7,8,9,10,11]. Of note, some reference laboratories have reported rates of carbapenem resistance *Bacteroides fragilis* were up to 7%, whereas resistance to antibiotics has been reported to vary by individual species and geographic regions [12,13,14]. These bacteria are associated with high mortality rates and are difficult to treat because of their high levels of antibacterial resistance and the lack of effective antibiotic regimens [15,16].

Under these circumstances, clinicians are increasingly using antibiotic combination regimens to treat infected patients. The breakpoint checkerboard assay is one of the methods for selecting optimal antibiotics against multidrug-resistant bacteria [17]. Nakamura et al. [18] reported that the clinical cure rate was 83.3% with antibiotic combination therapy according to the breakpoint checkerboard assay for multidrug-resistant *Pseudomonas aeruginosa* infections.

Some previous studies have suggested that combination regimens including clindamycin (CLDM) and minocycline (MINO) are effective against certain resistant anaerobes, such as the *B. fragilis* group and *P. anaerobius* [15,17,18,19,20]. Additionally, surveillance studies have demonstrated that carbapenems remain active against the great majority of species within the *B. fragilis* group [21,22,23,24]; thus, carbapenems still play an important role in the development of effective antibiotic combination regimens against multidrug-resistant anaerobes [25].

Nevertheless, few studies have evaluated the antibiotic effect of combination regimens of carbapenems with CLDM or MINO [26,27]. Therefore, this study aimed to investigate the in vitro efficacy of combination regimens of carbapenems with CLDM and MINO against *B. fragilis* and *P. anaerobius*.

## 2. Results

### 2.1. Antibiotic Susceptibilities of Study Isolates

Table 1 presents the in vitro susceptibility profiles of carbapenems, CLDM, and MINO against targeted anaerobic bacteria. This study included 18 *B. fragilis* isolates, 20 *B. thetaiotaomicron* isolates, 20 *Parabacteroides distasonis* isolates, and 12 *P. anaerobius* isolates. For *B. fragilis*, *Bacteroides thetaiotaomicron*, and *P. distasonis* isolates, minimum inhibitory concentration (MIC)90 and MIC50 values of imipenem (IPM), meropenem (MEPM), doripenem (DRPM), biapenem (BIPM), and panipenem (PAPM) were ≤2 and 0.75 μg/mL, respectively. For *P. anaerobius*, MIC90 and MIC50 values of IPM, MEPM, DRPM, BIPM, and PAPM were ≤1 and 0.5 μg/mL, respectively. For all study isolates, MIC90 and MIC50 of CLDM were >8 and ≤6 μg/mL, respectively. The MIC90 and MIC50 of MINO were 4 (except for *B. fragilis,* >8 μg/mL) and 2 μg/mL, respectively. 

### 2.2. Fractional Inhibitory Concentration Index (FICI) of Carbapenems and Clindamycin

Table 2 demonstrates the FICIs for targeted anaerobic bacteria, and Appendix A shows the MIC distribution in combination regimens with carbapenems and CLDM. For *B. fragilis*, each combination regimen displayed synergistic or additive effects in 33.3–38.8% of the study isolates, whereas MEPM + CLDM and other carbapenems (IPM, DRPM, BIPM, and PAPM) + CLDM showed antagonistic effects to 16.7% and 5.6% of the study isolates. For *B. thetaiotaomicron*, each combination regimen indicated synergistic or additive effects in 20.0–60.0% of the study isolates. Furthermore, IPM + CLDM indicated additive or synergistic effects in more than half of the study isolates (60.0%) without showing antagonistic effects. However, the other combination regimens, including MEPM, DRPM, BIPM, and PAPM, indicated antagonistic effects to 5.0–15.0% of study isolates. For *P. distasonis*, each combination regimen showed additive effects in 25.0–60.0% of the study isolates. IPM + CLDM displayed additive effects in more than half of the study isolates (60.0%) without showing antagonistic effects. However, the other combination regimens, including MEPM, DRPM, BIPM, and PAPM, showed antagonistic effects in 5–15.0% of the study isolates. For *P. anaerobius*, each combination regimen indicated synergistic or additive effects in 83.3–100.0% of the study isolates without indicating any antagonistic effects. IPM + CLDM and MEPM + CLDM indicated synergistic or additive effects in all study isolates (100.0%). 

### 2.3. FICI of Carbapenems and Minocycline

Table 3 shows the FICIs for targeted anaerobic bacteria, and Appendix A shows the MIC distribution in combination regimens with carbapenems and MINO. For *B. fragilis*, each combination regimen showed synergistic or additive effects in 33.3–72.2% of the study isolates, whereas IPM + MINO, BIPM + MINO, and PAPM + MINO exhibited antagonistic effects to 5.6%, 11.1%, and 5.6% of the study isolates, respectively. Moreover, IPM + MINO showed the highest synergistic and additive effects in the study isolates (72.2%). For *B. thetaiotaomicron*, each combination regimen showed additive effects to 20.0–75.0% of the study isolates. No combination regimen showed antagonistic effects, except PAPM + MINO (5.0%). IPM + CLDM and BIPM + MINO exhibited additive effects in more than half of the study isolates (75.0–60.0%) without showing antagonistic effects. For *P. distasonis*, each combination regimen indicated synergistic or additive effects in 30.0–80.0% of the study isolates, and no combination regimen indicated antagonistic effects. IPM + MINO showed additive effects in more than half of the study isolates (80.0%) without showing antagonistic effects. For *P. anaerobius*, each combination regimen displayed synergistic or additive effects in 91.7–100.0% of the study isolates without indicating any antagonistic effects. All combination regimens showed synergistic or additive effects in all study isolates (100.0%), except MEPM + MINO (91.7%).

## 3. Discussion

Several reports have indicated that multidrug-resistant anaerobic bacteria were detected among clinical isolates [12,13,14]. This study evaluated the antibiotic activities of combination regimens, including carbapenems (IPM, MEPM, DRPM, BIPM, and PAPM) and CLDM or MINO against anaerobic bacteria using the checkerboard assay. Consequently, these combination regimens showed synergistic or additive antibiotic effects in the majority (>80.0%) of *P. anaerobius* isolates. In contrast, the same combination regimens did not show remarkable synergistic effects against some *Bacteroides* species, such as *B. fragilis*, *B. thetaiotaomicron*, and *P. distasonis* (Table 2). 

To the best of our knowledge, this is the first study to demonstrate the antibiotic activities of combination regimens, including carbapenems against anaerobic bacteria. CLDM has been commonly used to treat some anaerobic bacterial infections, and MINO has been widely used to treat aerobic and anaerobic bacterial infections, such as odontogenic infections [16]. In clinical practice, IPM is mixed with cilastatin, which works as dehydropeptidase-1 (DHP-1 inhibitor, to prevent IPM degradation by DHP-1 [28]. PAPM was mixed with betamipron (BP), which acts as an organic anion transport inhibitor to reduce renal toxicity [21]. However, in our study, we did not add DHP-1 and BP to evaluate the antibiotic effects of combination regimens, including IPM and PAPM, because these inhibitors have a less antibacterial effect [29,30]. 

Compared with each single carbapenem regimen, the combination regimens of each carbapenem with CLDM did not enhance antibiotic activities against *B. fragilis* group species in our in vitro study. For most study isolates, FICIs of combination regimens were classified as additive or indifferent effects (Table 2). Kato et al. [22] reported that combination regimens of penicillins and cephalosporins with CLDM showed synergistic effects against *B. fragilis*. However, we observed only a few synergistic effects for the *B. fragilis* group isolates with combination regimens, including each carbapenem and CLDM. 

Similar to *B. fragilis*, combination regimens of each carbapenem and CLDM did not cause remarkable MIC changes in *B. thetaiotaomicron* and *P. distasonis* (Appendix A). However, FICIs of the combination regimen including IPM + CLDM showed a synergistic or additive effect against the 60% *B. thetaiotaomicron* strains we used. Notably, in a recent report, 8.2% of *B. thetaiotaomicron* strains were confirmed to be intermediate or resistant to IPM [31], and *B. fragilis* and *B. thetaiotaomicron* accounted for a slight share of the observed resistance to IPM and CLDM [32,33]. Hence, a combination regimen of IPM + CLDM can be an option to treat infections caused by multidrug-resistant *B. thetaiotaomicron*. 

Furthermore, in combination regimens with each carbapenem and CLDM against *P. anaerobius*, FICIs of almost all strains showed synergistic or additive effects (Table 3). Brook et al. [34] reported that penicillins combined with CLDM against anaerobic gram-positive cocci, including *P. anaerobius*, indicating synergistic or additive effects. Similar to this report, our study results suggest that combination regimens with carbapenems and CLDM have synergistic or additive effects against anaerobic gram-positive cocci (Table 3). Hence, these antibiotic combinations can be an option to treat infections caused by *P. anaerobius* isolates.

In contrast, although the antibiotic effect of combination regimens with β-lactams and MINO has already been evaluated against aerobic bacteria [35], few studies have evaluated them against anaerobic bacteria. In combination with each carbapenem and MINO, except IPM + MINO against *B. fragilis*, *B. thetaiotaomicron*, and *P. distasonis*, we did not observe remarkable MIC changes between single carbapenem regimens and each combination regimen (Appendix A–C). In total, 50–70% of the isolates were classified into an indifferent effect with FICIs. Only IPM + MINO showed synergistic or additive effects against *B. fragilis* group species, but the exact mechanism is unknown. 

Compared with single carbapenem regimens, combination regimens with each carbapenem and MINO showed enhanced antibiotic effects on *P. anaerobius*. The combination regimen resulted in a lower MIC. FICIs were indicated to have synergistic or additive effects in almost all strains. However, the number of isolates used in this study was limited. Hence, although MINO is expected to have a synergistic effect against gram-positive anaerobic isolates, more detailed research using a large number of samples is needed.

Our current study had some limitations. First, the number of clinical isolates evaluated in this in vitro study was small. Related to the limitations, our study results suggested combination regimens of each carbapenem with CLDM or MINO showed not only synergistic or additive effects but also antagonistic effects against the *B. fragilis* group (5.0–16.7%). Hence, further research is needed with more isolates to reveal the reason why the same strains showed variate combination effects. Second, our study evaluated the antibiotic activities of some combination regimens only with an in vitro study using checkerboard assay. Although a previous study reported that the results of the checkerboard assay were correlated with clinical response [18], this method does not consider the actual drug transfer to the local infected lesion and the pharmacokinetics of each drug. Additionally, carbapenems change their antibacterial activity under pH conditions [36]. Hence, various paternal in vivo research, such as a rat model of an abscess, is needed to validate our in vitro results. Third, we admitted the reduction in MICs, for *Bacteroides* spp., is mostly one dilution, and this may not provide enough reduction to achieve pharmacokinetics/pharmacodynamics targets for more resistant isolates. Finally, the carbapenem concentration evaluated in this study was within the susceptible ranges, and the study isolates we used are limited. Hence, we could not evaluate synergy/antagonism discussion at different MICs tested. 

However, the effectiveness of antibiotic combination therapy against pathogenic anaerobes has remained unclear, while some guidelines recommend using combination regimens to treat the infections due to multi-antibiotic resistant aerobic pathogens. We are thinking that one of the reasons is that the data evaluating the effectiveness against anaerobes have been extremely lacking, compared with aerobes. Additionally, reports of multidrug-resistant bacteria have been increasing in anaerobic bacteria. Therefore, our study results would be useful data to explore the preferable antibiotic combination regimens to show enhanced antibiotic activity to clinically important pathogenic anaerobes as preliminary results on in vitro activity of the antibiotic combinations with anti-anaerobic activity. Also, we believe that our results can provide new insights to explore the effective combination regimens against multidrug-resistant anaerobic bacteria as empirical and definitive therapies, while further studies are needed against highly antibiotic-resistant anaerobic isolates to carbapenems. 

## 4. Materials and Methods

### 4.1. Antibiotics and Checkerboard Production

We selected IPM, MEPM, PAPM, DRPM, and BIPM as carbapenems and CLDM and MINO. We evaluated the antibiotic efficacy of combination regimens with each carbapenem, CLDM, or MINO. Checkerboard plates were purchased from Eiken Chemical Co., Ltd. (Tokyo, Japan). The plates were 96-well microtiter plates and cation-adjusted Brucella broth for susceptibility testing. As shown in Figure 1, each plate comprised of a drug-free medium as a control, a medium containing a single drug to measure MICs, and a medium containing two drugs to evaluate the combined efficacy.

### 4.2. Bacterial Strains

*B. fragilis*, *B. thetaiotaomicron*, *P. distasonis*, and *P. anaerobius* from clinical isolates were kindly provided by the Gifu University Hospital, Japan. All strains were frozen at −80 °C with 15% skim milk suspension. To evaluate the antibiotic effects of combination regimens with carbapenems and CLDM or MINO, 18 strains of *B. fragilis*, 20 strains of *B. thetaiotaomicron*, 20 strains of *P. distasonis*, and 12 strains of *P. anaerobius* were tested.

### 4.3. Antibiotic Susceptibility Test

Anaerobic strains were cultured in anaerobic medium gum broth (Nissui Pharmaceutical Inc., Tokyo, Japan) for 24–48 h at 35 ± 1 °C. The strains were suspended turbidimetrically at 105 colony forming units (CFU)/well using anaerobic bacterial culture medium (ABCM) broth and ABCM bouillon broth (Eiken Chemical Co., Ltd., Tokyo, Japan). Drug susceptibility tests were demonstrated by broth microdilution method for 46–48 h at 36 ± 1 °C, according to the Clinical and Laboratory Standards Institute method [37]. The susceptible MIC break points of the CLSI criteria were as below IPM; 4 µg/mL, MEPM; 4 µg/mL, DRPM; 2 µg/mL, CLDM; 2 µg/mL and PAPM, BIPM and MINO were unspecified). The anaerobic culture apparatus used was an anaerobic box (Hirasawa Works Co., Ltd., Tokyo, Japan).

### 4.4. Evaluation of Antibiotic Combination Effect

To evaluate the antibiotic efficacy of combination regimens with each carbapenem and CLDM or MINO, FICI was calculated as follows: FICI = FIC of carbapenems + FIC of CLDM or MINO, where FIC of carbapenems (with/without CLDM or MINO) was defined as the ratio of the MICs of carbapenems (with/without CLDM or MINO) in combination, and the MIC of carbapenems (with/without CLDM or MINO) alone. The FICI values were interpreted as follows: ≤0.5, synergistic; >0.5, <2.0, additive; >1.0–≤2.0, indifferent; and >2.0, antagonistic effects [38].

## 5. Conclusions

This study evaluated the antibiotic activities of combination regimens including carbapenems and CLDM or MINO against anaerobic bacteria using checkerboard assay. Combination regimens of each carbapenem and CLDM or MINO indicated synergistic or additive effects against *P. anaerobius*. Although IPM plus MINO showed mainly additive effects against *B. fragilis*, *B. thetaiotaomicron*, and *P. distasonis*, the other combination regimens including MEPM, DRPM, BIPM, and PAPM with CLDM or MINO showed mainly indifferent effects on the anaerobic isolates in our study. Our results have some potential to provide useful information against future antibiotic-resistant anaerobic strains.

## Figures and Tables

**Figure 1 antibiotics-11-00292-f001:**
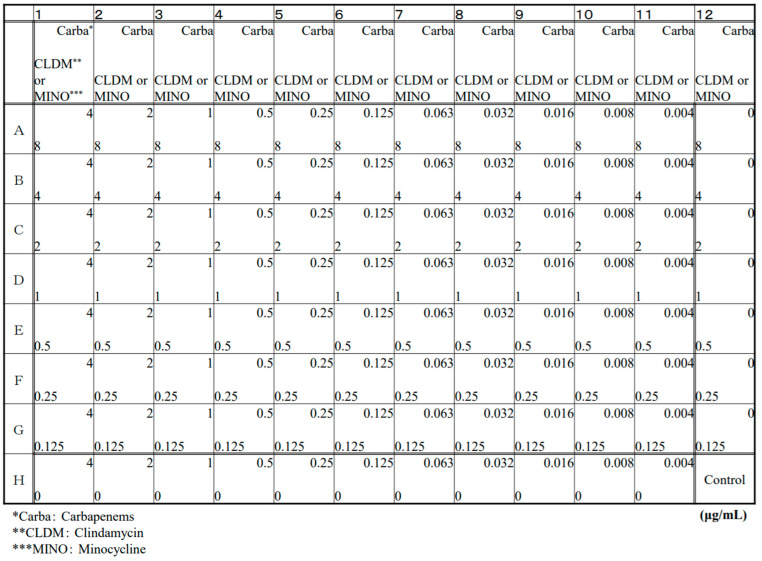
Drug concentration of carbapenems and clindamycin or minocycline by checkerboard assay.

**Table 1 antibiotics-11-00292-t001:** Drug susceptibility testing of anaerobic bacteria.

		MIC (μg/mL)		
		Range	MIC50	MIC90
*B. fragilis*	(*n* = 18)			
IPM		0.063–1	0.125	0.5
MEPM		0.008–1	0.125	1
DRPM		0.032–1	0.125	1
BIPM		0.032–2	0.125	1
PAPM		0.063–2	0.25	2
CLDM		0.25–8<	0.75	8<
MINO		1–8<	2	8<
*B. thetaiotaomicron*	(*n* = 20)			
IPM		0.25–1	0.5	1
MEPM		0.032–1	0.125	0.25
DRPM		0.125–1	0.5	0.5
BIPM		0.06–0.5	0.25	0.5
PAPM		0.25–2	0.5	2
CLDM		1–8<	6	8<
MINO		1–4	2	4
*P. distasonis*	(*n* = 20)			
IPM		0.5–2	1	2
MEPM		0.25–2	0.75	2
DRPM		0.063–2	0.5	2
BIPM		0.032–1	0.125	0.5
PAPM		0.03–2	0.125	2
CLDM		0.25–8<	4	8<
MINO		1–8<	2	4
*P. anaerobius*	(*n* = 12)			
IPM		0.03–1	0.125	1
MEPM		0.015–1	0.125	0.5
DRPM		0.06–1	0.5	1
BIPM		0.03–1	0.125	0.5
PAPM		0.12–2	0.5	1
CLDM		0.25–8<	0.5	8<
MINO		1–8<	2	4

**Table 2 antibiotics-11-00292-t002:** Fractional inhibitory concentration index (FICI) for anaerobic bacteria in combination with carbapenems plus clindamycin.

	FICI			
	≤0.5 Synergistic	>0.5 to ≤1.0 Additive	>1.0 to ≤2.0 Indifferent	>2.0 Antagonistic
*B. fragilis* (*n* = 18)				
Imipenem	0	6 (33.3%)	11 (61.1%)	1 (5.6%)
Meropenem	0	6 (33.3%)	9 (50.0%)	3 (16.7%)
Doripenem	3 (16.6%)	4 (22.2%)	10 (55.6%)	1 (5.6%)
Biapenem	0	7 (38.8%)	10 (55.6%)	1 (5.6%)
Panipenem	0	7 (38.8%)	10 (55.6%)	1 (5.6%)
*B. thetaiotaomicron* (*n* = 20)				
Imipenem	1 (5.0%)	11 (55.0%)	8 (40.0%)	0
Meropenem	0	4 (20.0%)	15 (75.0%)	1 (5.0%)
Doripenem	0	6 (30.0%)	11 (55.0%)	3 (15.0%)
Biapenem	0	10 (50.0%)	8 (40.0%)	2 (10.0%)
Panipenem	0	6 (30.0%)	13 (65.0%)	1 (5.0%)
*P. distasonis* (*n* = 20)				
Imipenem	0	12 (60.0%)	8 (40.0%)	0
Meropenem	0	10 (50.0%)	9 (45.0%)	1 (5.0%)
Doripenem	0	5 (25.0%)	12 (60.0%)	3 (15.0%)
Biapenem	0	10 (50.0%)	8 (40.0%)	2 (10.0%)
Panipenem	0	5 (25.0%)	14 (70.0%)	1 (5.0%)
*P. anaerobius* (*n* = 12)				
Imipenem	4 (33.3%)	8 (66.7%)	0	0
Meropenem	4 (33.3%)	8 (66.7%)	0	0
Doripenem	6 (50.0%)	4 (33.3%)	2 (16.7%)	0
Biapenem	7 (58.4%)	4 (33.3%)	1 (8.3%)	0
Panipenem	2 (16.7%)	8 (66.6%)	2 (16.7%)	0

**Table 3 antibiotics-11-00292-t003:** FICI for anaerobic bacteria in combination with carbapenems plus minocycline.

	FICI			
	≤0.5 Synergistic	>0.5 to ≤1.0 Additive	>1.0 to ≤2.0 Indifferent	>2.0 Antagonistic
*B. fragilis* (*n* = 18)				
Imipenem	1 (5.6%)	12 (66.6%)	4 (22.2%)	1 (5.6%)
Meropenem	0	7 (38.9%)	11 (61.1%)	0
Doripenem	0	6 (33.3%)	12 (66.7%)	0
Biapenem	0	7 (38.9%)	9 (50.0%)	2 (11.1%)
Panipenem	0	7 (38.9%)	10 (55.5%)	1 (5.6%)
*B. thetaiotaomicron* (*n* = 20)				
Imipenem	0	15 (75.0%)	5 (25.0%)	0
Meropenem	0	4 (20.0%)	16 (80.0%)	0
Doripenem	0	5 (25.0%)	15 (75.0%)	0
Biapenem	0	12 (60.0%)	8 (40.0%)	0
Panipenem	0	6 (30.0%)	13 (65.0%)	1 (5.0%)
*P. distasonis* (*n* = 20)				
Imipenem	0	16 (80.0%)	4 (20.0%)	0
Meropenem	0	6 (30.0%)	14 (70.0%)	0
Doripenem	0	7 (35.0%)	13 (65.0%)	0
Biapenem	0	6 (30.0%)	14 (70.0%)	0
Panipenem	1 (5.0%)	6 (30.0%)	13 (65.0%)	0
*P. anaerobius* (*n* = 12)				
Imipenem	4 (33.3%)	8 (66.7%)	0	0
Meropenem	4 (33.3%)	7 (58.3%)	1 (8.3%)	0
Doripenem	1 (8.3%)	11 (91.7%)	0	0
Biapenem	0	12 (100.0%)	0	0
Panipenem	1 (8.3%)	11 (91.7%)	0	0

## Data Availability

The data from the current study can be provided upon reasonable request from the corresponding author.

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
