# Peer review of "In Vitro Efficacy of Antibiotic Combinations with Carbapenems and Other Agents against Anaerobic Bacteria"

_antibiotics, 2022, doi:10.3390/antibiotics11030292_

Round 1

Reviewer 1 Report

The paper is well writen and concise. There are not many studies addressing antimicrobial resistance of anaerobes, and even less so the synergistic effect of drugs with anaerobic activity.

There are several limitations of the study. The first one, that have been mentioned by the authors already, is the small number of isolates. The other less understandabe limitation, that is also mentioned, is the choice of isolates that are susceptible to carbapenems. Since the isolates were not collected prospectively, the authors could probably use resistant isolates from the bank with frozen bacteria. That makes the study much less relevant, since the combination is not needed and is in fact undesirable when there are effective drugs available.

In spite of original idea of the study, the study is too modest for publication.

Here are a few comments that could help improve the paper in the case of submission to another journal.

The authors should certainly put the study in the perspective. It brings preliminary results on in vitro activity of the combinations of antibiotics with anti-anaerobic activity, but the results should not point to the use of combination when isolates are susceptible to monotherapy. Besides, by the authors' opinion: what is the role for combination therapy of infections with anaerobes? To increase the efficacy of antibiotics that are less effecive? As empirical treatment? The authors should provide a short perspecitve on clinical value of their findings.

Abstract

Several combinations are antagonistic, there are more frequent than synergistic for most bacteria included in the study. The authors should mention this in the abstract in a balanced way. Introduction

Line 29: references 1 and 2 are outdated and should be replaced by some more recent ones, also describing the role of immune insufficiency in infections with anaerobes. This relationship is not so clear.

Line 51: the authors state that there are few studies: they should be referenced

Discussion

Some combinations showed to be antagonistic. The authors should stress this in the discussion.

Line 199: what do the authors mean with the paternal studies?

Methods

5.2. Where the isolates stored in a bank or these are prospecively collected isolates from clinical specimens – which specimens?

Author Response

# Reviewer 1

The authors should certainly put the study in the perspective. It brings preliminary results on in vitro activity of the combinations of antibiotics with anti-anaerobic activity, but the results should not point to the use of combination when isolates are susceptible to monotherapy. Besides, by the authors' opinion: what is the role for combination therapy of infections with anaerobes? To increase the efficacy of antibiotics that are less effective? As empirical treatment? The authors should provide a short perspective on clinical value of their findings.

RESPONSE: Thank you. As reviewer pointed out, the effectiveness of antibiotic combination therapy against pathogenic anaerobes has remain unclear, while some guidelines recommended to use combination regimens to treat the infections due to multi-antibiotic resistant aerobic pathogens. We are thinking the one of the reasons is the data evaluating the effectiveness against anaerobes have been extremely lacking, compared with aerobes. Additionally, as mentioned in introduction, reports of multidrug-resistant bacteria have been increasing in anaerobic bacteria. Therefore, our study results would be useful data to explore the preferable antibiotic combination regimens to show enhanced antibiotic activity to clinically important pathogenic anaerobes as preliminary results on in vitro activity of the antibiotic combinations with anti-anaerobic activity. We believe that our results can provide new insights to explore the effective combination regimens against multidrug-resistant anaerobic bacteria as empirical and definitive therapies. These sentences added in limitation section. (Line 212-224)

Abstract

Several combinations are antagonistic, there are more frequent than synergistic for most bacteria included in the study. The authors should mention this in the abstract in a balanced way.

RESPONSE: Thank you for your advice. We revised our abstract, and added sentences regard to the antagonistic results.

Introduction

Line 29: references 1 and 2 are outdated and should be replaced by some more recent ones, also describing the role of immune insufficiency in infections with anaerobes. This relationship is not so clear.

RESPONSE: Thank you for your pointed out. We changed the references 1 and 2 to update. Reference 1 indicates immune insufficiency can cause opportunistic infections including infections with anaerobes and, then, underlying diseases (e.g. hemodialysis, malignancy and diabetes) are risk factors for anaerobic bacteremia. (Line 30-32)

Line 51: the authors state that there are few studies: they should be referenced.

RESPONSE: Thank you for your pointed out. We added the references. (Line 56)

Discussion

Some combinations showed to be antagonistic. The authors should stress this in the discussion.

RESPONSE: Thank you for your advice. We added the sentences shown below in limitation section Line 195-199.

“Our current study had some limitations. First, the number of clinical isolates evaluated in this in vitro study was small. Related to the limitation, our study results suggested combination regimens of each carbapenem with CLDM or MINO showed not only syner-gistic or additive effects, but also antagonistic effects against and B. fragilis group (5.0-16.7%). Hence, further research is needed with more isolates to reveal the reason why same species showed variate combination effects.”

Line 199: what do the authors mean with the paternal studies?

RESPONSE: Thank you for your pointed out. Carbapenems change their antibacterial activity under pH conditions such as abscess. We added sentences Line 205-208.

Methods

5.2. Where the isolates stored in a bank or these are prospectively collected isolates from clinical specimens – which specimens?

RESPONSE: Study isolates were kindly provided from the department of microbiology in Gifu University Hospital, and they were mainly isolated from blood and abscess. However, we are terribly sorry for that there's nothing left to determine where some isolates detected.

Reviewer 2 Report

This is a well designed study with clear results and I have few comments.

The only major issue with the study is the choice of antibiotic susceptible isolates to undertake synergy testing with.  There is no clinical requirement to achieve synergy for these bacteria and although the authors acknowledge the utility will be in the setting of multi-drug resistance the results cannot necessarily be extrapolated.   The reduction in MICs, for Bacteroides spp., is modest at mostly 1 dilution and this may not provide sufficient reduction to achieve PKPD targets for more resistant isolates.  I suggest the authors elaborate on this in the discussion.  It would be interesting to consider the synergy/antagonism discussion at different MICs tested.

Minor points to consider:

  1. Line 34.  Resistance rates quoted at 7%.  Resistance to what?
  2. Please include CLSI interpretative criteria to more clearly distinguish these isolates as resistant/susceptible.
  3. Please describe when clinical isolates were collected from and why the number of strains were chosen.

Author Response

# Reviewer 2

The only major issue with the study is the choice of antibiotic susceptible isolates to undertake synergy testing with. There is no clinical requirement to achieve synergy for these bacteria and although the authors acknowledge the utility will be in the setting of multi-drug resistance the results cannot necessarily be extrapolated. 

RESPONSE: Thank you. As reviewer pointed out, the effectiveness of antibiotic combination therapy against pathogenic anaerobes has remain unclear, while some guidelines recommended to use combination regimens to treat the infections due to multi-antibiotic resistant aerobic pathogens. We are thinking the one of the reasons is the data evaluating the effectiveness against anaerobes have been extremely lacking, compared with aerobes. Additionally, as mentioned in introduction, reports of multidrug-resistant bacteria have been increasing in anaerobic bacteria. Therefore, our study results would be useful data to explore the preferable antibiotic combination regimens to show enhanced the antibiotic activity to clinically important pathogenic anaerobes as preliminary results on in vitro activity of the antibiotic combinations with anti-anaerobic activity. These sentences added in limitation section. (Line 212-224)

The reduction in MICs, for Bacteroides spp., is modest at mostly 1 dilution and this may not provide sufficient reduction to achieve PK/PD targets for more resistant isolates. I suggest the authors elaborate on this in the discussion. It would be interesting to consider the synergy/antagonism discussion at different MICs tested.

RESPONSE: Thank you. As you suggested, we added following sentences as our study limitation in the discussion part (Line 206-208). “Third, we admitted the reduction in MICs, for Bacteroides spp., is mostly one dilution and this may not provide enough reduction to achieve pharmacokinetics/pharmacodynamics targets for more resistant isolates.”

Additionally, as we mentioned in limitation part, study isolates we used are limited and are susceptible to carbapenems. Hence, we could not evaluate synergy/antagonism discussion at different MICs tested. (Line 208-211)

Minor points to consider:

  1. Line 34.  Resistance rates quoted at 7%.  Resistance to what?

RESPONSE: Thank you for your pointed out. We added sentences Line 37.

  1. Please include CLSI interpretative criteria to more clearly distinguish these isolates as resistant/susceptible.

RESPONSE: Thank you for your advice. We added sentences Line 287-289.

  1. Please describe when clinical isolates were collected from and why the number of strains were chosen.

RESPONSE: Study isolates were kindly provided from the department of microbiology in Gifu University Hospital, and they were mainly isolated from blood and abscess. However, we are terribly sorry for that there's nothing left to determine where some isolates detected. Additionally, the reason why we chose the number of strains is based on the number of strains available. The limited number of isolates is discussed in limitation section (Line 194-195).

Round 2

Reviewer 1 Report

The authors have addressed appropriately all the concerns.

Author Response

Thank you.